# Molecular snapshot of drug-resistant *Mycobacterium tuberculosis* strains from the Plateau State, Nigeria

Zofia Bakuła[1☯], Valentine B. Wuyep[2☯], Łukasz Bartocha[1], Anna Vyazovaya[3], Eugene I. Ikeh[4], Jacek Bielecki[1], Igor Mokrousov[3]*, Tomasz Jagielski[1]*

**1** Department of Medical Microbiology, Institute of Microbiology, Faculty of Biology, University of Warsaw, Warsaw, Poland, **2** Plateau State College of Health Technology, Zawan, Jos, Nigeria, **3** Laboratory of Molecular Epidemiology and Evolutionary Genetics, St. Petersburg Pasteur Institute, St. Petersburg, Russia, **4** Department of Medical Microbiology, School of Medical and Health Sciences, College of Medical Sciences, University of Jos, Jos, Nigeria

☯ These authors contributed equally to this work.

* imokrousov@mail.ru (IM); t.jagielski@biol.uw.edu.pl (TJ)

**Data Availability Statement:** All relevant data are within the manuscript and its Supporting Information files.

## Abstract

Nigeria ranks 1st in Africa and 6th globally with the highest burden of tuberculosis (TB). However, only a relatively few studies have addressed the molecular epidemiology of *Mycobacterium tuberculosis* in this country. The aim of this work was to analyze the genetic structure of drug-resistant (DR) *M. tuberculosis* population in the Plateau State (central Nigeria), with the results placed in the broader context of West Africa. The study sample included 67 DR *M. tuberculosis* isolates, recovered from as many TB patients between November 2015 and January 2016, in the Plateau State. The isolates were subjected to spoligotyping and MIRU-VNTR typing. A total of 20 distinct spoligotypes were obtained, split into 3 clusters (n = 50, 74.6%, 2–33 isolates per cluster) and 17 (25.4%) unique patterns. The Cameroon clade was the largest lineage (62.7%) followed by T (28.3%), LAM (3%), and Haarlem (3%) clades. Upon MIRU-VNTR typing, the isolates produced 31 profiles, i.e. 7 clusters (n = 43, 64.2%, 2–17 isolates per cluster) and 24 singletons. A combined spoligotyping and MIRU-VNTR typing analysis showed 20.9% of the cases clustered and estimated the recent transmission rate at 11.9%. In conclusion, two lineages, namely Cameroon, and T accounted for the majority (91%) of cases. No association was observed between the most prevalent Cameroon lineage and drug resistance, including multidrug resistant (MDR) phenotype, or any of the patient demographic characteristics.

## Introduction

Tuberculosis (TB) in Africa has a long history of counteracting and overlapping events. Anatomically modern humans have originated in the easternmost region of the African continent and so could be the case for a causative agent of human TB. The discovery of the increasing number of *Mycobacterium tuberculosis* lineages restricted to East Africa supports this

**Funding:** The study was in part supported by the University of Warsaw's intramural grants: for the employees' research potential increase (no. BOB-661-98/2019) and for young investigators (no. 501-D114-01-1140400).

**Competing interests:** The authors have declared that no competing interests exist.

notion [1]. Nonetheless, most of the globally prevalent *M. tuberculosis* lineages have originated, evolved, and adapted to local human populations on other continents and only historically recently have been brought back to the African continent. This is especially the case of the Euro-American Lineage (Lineage 4), which had hypothetically emerged a few millennia ago in the Eurasian heartland and subsequently spread in all directions but mainly westwards [2]. The ancestral Lineage 4 types were then brought to most of Africa in the 18th and 19th centuries, and started to evolve locally thereafter. Consequently, several genetic lineages were identified and named after their countries of first isolation, such as Uganda, Ghana, Cameroon etc. They were initially proposed upon distinct spoligotyping and Variable Number of Tandem Repeats (VNTR) typing patterns. More recently, the whole-genome sequencing (WGS)-based studies confirmed their identity and supported the existence of more clonal lines within Lineage 4, such as the Congo [3]. Unlike autochthonous *Mycobacterium africanum*, first described in Senegal in 1968 [4], newly imported strains of *M. tuberculosis* demonstrated increased pathogenic capacities against the naïve local population in the sub-Saharan Africa. This seemingly contrasts with a belief that a history of TB exposure provides resistance to the disease. It appears, however, that current *M. tuberculosis* strains circulating at least in West Africa descended from the Euro-American lineage, which followed a historical local evolution influenced by adaptation to humans and occasional human migration [1]. Thus, the higher susceptibility of black populations, compared to Caucasian people, to modern TB clades can be explained by the fact that TB has been endemic in Europe for a much longer period [5, 6].

Nigeria holds one of the top positions on the list of 30 high TB burden countries. It ranks 6th for the global TB crisis, with an estimated 440,000 new TB cases and 154,000 TB-related deaths in 2019 [7]. Nigeria is also among countries with the highest global multidrug resistant/rifampicin resistant (MDR/RR) TB and TB/HIV coinfection rates, approaching 11 and 23 per 100,000 population, respectively [7]. Noteworthy, the level of MDR-TB may be higher than officially reported [8].

Nigeria is marked with a high level of ethnic and language diversity. In particular, the Afro-Asiatic language family includes Hausa, Ngas, Goemai, and Mwaghavul languages (northern region), while Niger–Congo language family includes Berom, Yoruba, Tarok, and Tiv languages (southern and central regions). Islam is the predominant religion of the Hausa people, while Christianity and local religions are practiced in other ethnic groups. This ethnic and religious stratification may be worth taking into consideration when interpreting molecular typing data of *M. tuberculosis* and formulating epidemiological surveillance policies.

Molecular epidemiology of human TB in Nigeria has been investigated in a limited number of studies, mainly based on isolates collected in 2014 or before [8–14]. Most of these works have explored *M. tuberculosis* variability at a regional level [8–10, 12]. Only once has the genetic diversity of the Nigerian *M. tuberculosis* population been evaluated with a widely accepted combination of spoligotyping and 24-locus MIRU-VNTR typing [11]. All the remaining studies have used either spoligotyping only [8, 9, 13, 14] or spoligotyping in combination with a 5- or 12-locus MIRU-VNTR typing [10, 12]. The Cameroon lineage consistently predominated, and usually was followed by *M. africanum* [8, 10–13]. In central Nigeria, sublineage L4.6.2 (Cameroon linegae) was mostly identified [13]. Across the studies, the overall recent transmission of TB was high and accounted from 38% up to 61.1% of the cases, without major regional differences, and with clearly delineated MDR-TB clusters [10–12].

This study was largely motivated by the paucity of research on molecular epidemiology of *M. tuberculosis* in Nigeria. Its advantage lies in the combined use of spoligotyping and MIRU-VNTR typing on drug-resistant (DR) *M. tuberculosis*. Moreover, for the first time, the association between TB spoligotype signatures and patients' ethnicity and language groups in

Nigeria was determined. Additionally, the obtained phylogenetic results were placed in a wider spatiotemporal context of West Africa.

## Materials and methods

### Study sample

The study sample included 67 *M. tuberculosis* isolates, resistant to at least one first-line drug, recovered from as many pulmonary TB patients admitted to three different hospitals in central Nigeria, over a 3-month period (November 2015-January 2016). The hospitals were: (i) Plateau State Specialist Hospital TB DOTs Center; (ii) Jos University Teaching Hospital MDR-TB Clinic; (iii) Ministry of Health TB Prevention and Control Center in Jos, Plateau State, and covered Plateau State. The total number of *M. tuberculosis* isolates collected in the study period was 45 for the Plateau State Specialist Hospital TB DOTs Center and 145 for Jos University Teaching Hospital MDR-TB Clinic. Data on the total *M. tuberculosis* isolates from the Ministry of Health TB Prevention and Control Center in Jos was unavailable. Also, not for all *M. tuberculosis* isolates were data on their drug resistance profiles obtainable.

Since no data on patient management and long-term treatment outcomes were available, they were excluded from the analysis.

Primary isolation, culturing, and species identification were performed with standard mycobacteriological methods [15].

All methods were carried out in accordance with the guidelines and recommendations of University of Jos Ethics Committee. All experimental protocols were approved by University of Jos Ethics Committee. All data were anonymized prior the study, therefore the need for consent was waived by the Ethics Committee.

### Drug susceptibility testing

Drug susceptibility testing (DST) for first-line drugs was performed using the standard 1% proportion method on the Löwenstein-Jensen medium, following the WHO protocols [16]. The *M. tuberculosis* H37Rv reference strain was used as a control. The critical concentrations for specific drugs were as follows: isoniazid (INH), 0.2 mg/L; rifampicin (RIF), 40 mg/L; ethambutol (EMB), 2 mg/L; streptomycin (STR), 4 mg/L.

### DNA isolation

Extraction of genomic *M. tuberculosis* DNA was done using the cetyl-trimethyl ammonium bromide (CTAB) method, as described elsewhere [17]. The purified DNA was dissolved in TE buffer (10 mM Tris-HCl, 1 mM EDTA, pH 8.0) and quantified with the NanoDropTM 2000 Spectrophotometer (ThermoFisher Scientific, Waltham, USA). The DNA samples were diluted to the required concentration (*ca*. 10 ng/μL) and stored at –20˚C until used.

### Spoligotyping

Spoligotyping was performed using commercial kits (Ocimum Biosolutions, India) and following the published protocol [18]. All profiles were assessed by two independent researchers. SITVIT2 database [19; http://www.pasteur-guadeloupe.fr:8081/SITVIT2/) was used to ascertain Spoligotype International Type (SIT) numbers for the isolates studied.

### MIRU-VNTR-typing

Genotyping at 24 MIRU-VNTR loci was performed, essentially as described previously [20] using agarose gels. MIRU-VNTR*plus* online resource (https://www.miru-vntrplus.org/) was

used for phylogenetic analysis of the MIRU-VNTR profiles. A PCR failure (i.e. multiple bands or no bands) was permanently observed for Mtub39 and QUB-11b. Consequently, those two loci were excluded from the analysis.

## Data analysis

A spoligotyping cluster was defined as two or more isolates sharing identical spoligotypes. The same criteria, i.e. exact match at 22 VNTR loci was applied for a MIRU-VNTR cluster. A minimum spanning tree was drawn based on the spoligotyping data with the SpolTools programme [21]. A dendrogram was constructed based on 22-locus MIRU-VNTR typing data, by using MIRU-VNTR*plus* software (https://www.miru-vntrplus.org/). The Hunter Gaston discriminatory index was calculated using the formula described earlier [22]. The Recent Transmission Index (RTI) was calculated using the "n-1" formula, i.e. $\frac{(N-C)}{SS}$, where $N$ is the number of clustered cases, $C$ is the number of clusters, and $SS$ is the sample size [23].

A chi-square test was used to detect any significant difference between the two groups. Yates corrected $\chi 2$ and *P*-values were calculated with 95% confidence interval at MEDCALC (http://www.medcalc.org/calc/odds_ratio.php) online resource.

# Results

## Socio-demographic characteristics

The mean age of 67 enrolled patients was 35.2±10.8 years (range 15–58). Male-to-female ratio was *ca*. 4:1. Three-fourths (77.6%) of the patients were from urban areas, whereas the others (22.4%) came from rural locations (S1 Table in S1 File). Most (77.6%) of the patients came from north part of the Plateau State (S1 Table in S1 File).

In general, patients belonged to eight ethnic groups, with main ethnicities being Hausa, Berom, and Ngas (S1 and S2 Tables in S1 File). Hausa and Berom patients (n = 48) were all from the northern region and all but 3 (93.7%) lived in urban areas. In contrast, patients from the central region mostly (90.9%) dwelled in rural areas and most (63.6%) of them belonged to the Ngas ethnic group (S1 and S2 Tables in S1 File).

## Drug susceptibility testing

Half (52.2%) of the isolates met the definition of MDR-TB. Among isolates resistant to more than one drug, the most common drug resistance profile was INH+RIF+STR+EMB (17.9%), followed by INH+RIF+EMB (16.4%), and INH+RIF (10.4%). Monoresistance was detected in 38.8% of the isolates, and mostly (80.8%) concerned STR (Table 1).

## Spoligotyping

In total, 20 spoligotyping profiles were identified, including 3 types shared by 2, 15, or 33 isolates, and 17 unique profiles. All 67 isolates belonged to the Lineage 4. The largest clusters were SIT61 (Cameroon) and SIT53 (T), represented by 33 (49.2%) and 15 (22.4%) isolates, respectively (Table 2). Five unique profiles could not be found in SITVIT2 and were thus designated as orphan/new types, although application of the SITVIT2 decision rules and expert assessment permitted to assign 3 such isolates to the Cameroon (n = 2) and T (n = 1) lineages. The remaining two isolates with undefined clonal line status were designated as L4-unclassified. Only two lineages (Cameroon and T) accounted for 91% of the isolates (62.7% and 28.3%, respectively).

A schematic view of the evolutionary relationships between spoligotypes is shown in Fig 1. The network was built based on the general knowledge of the evolution within Lineage 4, in

**Table 1. Drug susceptibility profiles of 67 isolates under the study.**

|  | DST profile* | No. of isolates (n/%) n = 67 |
|---|---|---|
| Non-MDR n = 32 (47.8%) | INH | 1/1.5 |
|  | RIF | 2/3 |
|  | EMB | 2/3 |
|  | STR | 21/31.3 |
|  | RIF+EMB | 2/3 |
|  | INH+EMB | 1/1.5 |
|  | INH+STR | 3/4.5 |
| MDR n = 35 (52.2%) | INH+RIF | 7/10.5 |
|  | INH+RIF+EMB | 11/16.4 |
|  | INH+RIF+STR | 5/7.4 |
|  | INH+RIF+EMB+STR | 12/17.9 |
|  | Total | 67/100 |

*isoniazid (INH), rifampicin (RIF), ethambutol (EMB), streptomycin (STR).

particular the ancestral state of the SIT53 and taking into consideration that *M. tuberculosis* CRISPR/DR locus is evolving by loss of spacers. Although the picture is simplified it gives an adequate view of the sequence of events behind evolution of the Nigerian spoligotypes, and a local historical evolution of the Cameroon lineage.

Two Haarlem strains had different SITs (Table 2) and due to their single deletion event difference, they could not be directly linked to each other. In addition, they differed in other

**Table 2. Spoligotype-based diversity of 67 *M. tuberculosis* isolates under the study.**

| SIT | Lineage | 43-signal spoligoprofile | No. of isolates (n/%) | |
|---|---|---|---|---|
| 61 | Cameroon | ■■■■■■■■■■■■■■■■■■■■■■■■□□□■■■■■■□□□□■■■■■■■ | 33/49.2 | 42/62.7 |
| 838 | | ■■■■■■■■■■■■■■■■■■■■■■□□□■■■■■■□□□□■■■■□■■ | 1/1.5 | |
| 2970 | | ■■■■■■■■■■■■■■■□■■■□□□■■■■■■■□□□□■■■■■■■ | 1/1.5 | |
| 3400 | | ■■■■■■■■■■■■■□□□□■■■□□□■■■■■■□□□□■■■■■■ | 1/1.5 | |
| 115 | | ■■■■■■■■■■■■■□■■■■■■□□□■■■■■■■□□□□■■■■■■■ | 1/1.5 | |
| 2550 | | ■■■■■■■■■■■□■■■■■■■■□□□■■■■■■□□□□■■■■■■ | 1/1.5 | |
| 852 | | ■□□□□□□□□□□□□□□□■■■■■■□□□■■■■■■■■□□□□■■■■■■■ | 1/1.5 | |
| 2830 | | ■■■■■■■■■■■■■■■■■■■□□□■□■■■■□□□□■■■■■■■ | 1/1.5 | |
| Orphan5new | | ■■■■■■■■■■■■■■□□■■■■□□□■■■■■■□□□□■■■■■■ | 1/1.5 | |
| Orphan1new | | ■■■■■■■■■■■■■■■■■■■■■■□□□■■■■■■■■□□□□■■■■□□□ | 1/1.5 | |
| 53 | T | ■■■■■■■■■■■■■■■■■■■■■■■■■□□□□□■■■■■■■ | 15/22.3 | 19/28.3 |
| 774 | | ■■■■■■■■■■■■■■■■■■■■■□■■■■■■□□□□□■■■■■■■ | 1/1.5 | |
| 1580 | | ■■■■■■■■■■■■■■■□□■■■■■■■□□□□■■■■■■■ | 1/1.5 | |
| 1913 | | ■■■■■■■■■■■■■■□■■■■■■■□■■■■■■■■□□□□■■■■■■ | 1/1.5 | |
| Orphan7new | | ■■■■■■■■■■■■■■□■■■■■■□□■■■■■■■□□□□■■■■■■■ | 1/1.5 | |
| 49 | Haarlem | ■■■■■■■■■■■■■■■■■■■■■■■■■■■■■■■□■□□□□■■■□■■■ | 1/1.5 | 2/3 |
| 316 | | ■■■■■■■■■■■■■■■■■■■■■■■■■■■■□□□□□□□■□□□□■■■□■■■ | 1/1.5 | |
| 17 | LAM | ■■□■■■■■■■■■□■■■■■■■□□□□■■■■■■■□□□□■■■■■■■ | 2/3 | 2/3 |
| Orphan8new | L4-unclass.* | ■■□■■■■■■■■■■■■□□■■■■■■■■■■■□■■□□□□■■■□■■■ | 1/1.5 | 2/3 |
| Orphan3new | | ■■■■■■■■■■■■■■■■■■■■■■□□□□□□□□□□□□□□□■■□□■■■ | 1/1.5 | |

*L4-unclass.– L4-unclassified; assignment to the Lineage in accordance with an expert assessment.

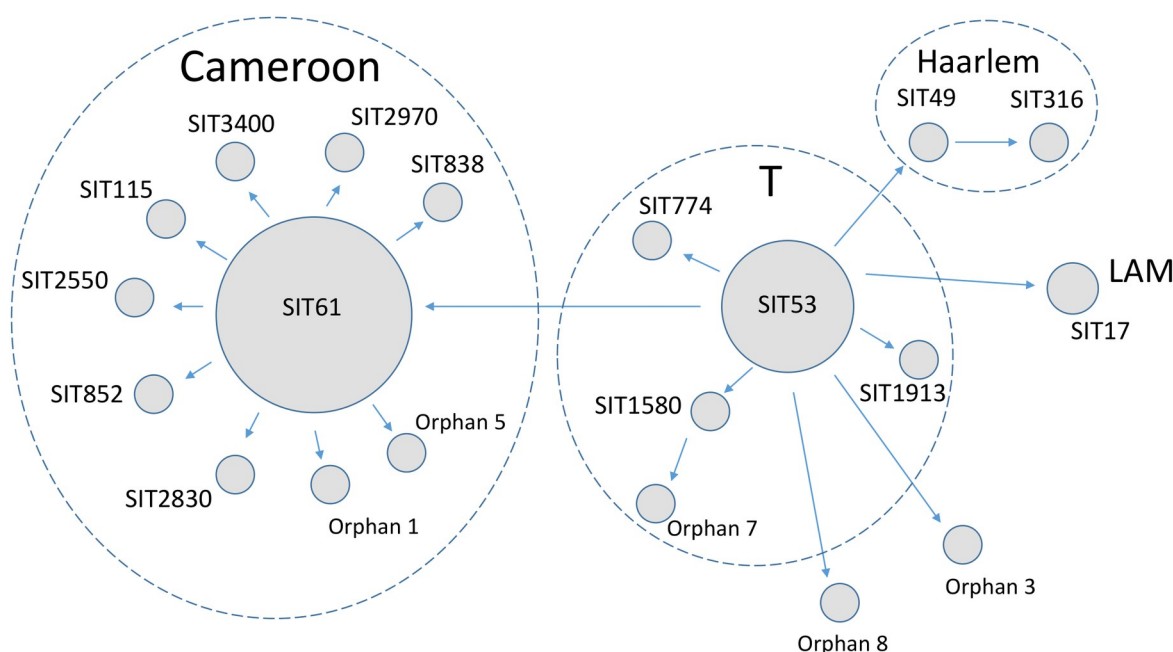

**Fig 1. Spoligoforest of 67 *M. tuberculosis* isolates from the Plateau State, Nigeria, illustrating the potential evolutionary relationships of the spoligotypes identified in this study.**

variables; hypothetically ancestral SIT49 was a STR-resistant strain, isolated from a Hausa patient based in the northern urban area, while SIT316 was a RIF-resistant strain from a Ngas patient dwelling in the central rural area. Therefore, these isolates could be at most, only distantly related. In contrast, LAM SIT17 isolates were both from patients living in urban northern area, yet of different ethnicities (Hausa and Ngas). The strains also exhibited different drug resistance profiles (INH+RIF+EMB *vs.* INH+RIF+STR).

Whereas the MDR rate was somewhat higher among T compared to Cameroon isolates (57.9% *vs.* 47.6%, *P* = 0.4), the proportion of MDR isolates among two major types, i.e. SIT61 and SIT53 was almost identical (45.5% *vs.* 46.7%) (S3 Table in S1 File).

Concerning the geographical distribution, the Cameroon lineage was more prevalent in the center of the Plateau State, compared to north (81.8% *vs.* 63.5%, *P* = 0.2), while T lineage was more common in the north (28.8% *vs.* 9.1%, P = 0.2). The northern sample was somewhat more diverse in terms of HGDI than the central one (S4 Table in S1 File). No significant difference was found between urban and rural subgroups, except for slightly higher prevalence of SIT61 in the urban setting and higher diversity in the rural setting (S5 Table in S1 File).

The MDR isolates were more frequent in the Berom than Hausa population (59.1% *vs.* 50%; P = 0.5) (S6 and S7 Tables in S1 File). The Cameroon genotypes were found at a similar frequency in the two ethnic groups, but the T lineage and SIT53 type were more common among the Berom people (*P* = 0.3 and 0.5, respectively).

In addition to the ethnic subdivision, the isolates were grouped according to the language family and religion (S7–S9 Tables in S1 File). The T clonal line was more prevalent in the Niger-Congo language family (38.5% *vs.* 22.0%; P = 0.1). Two LAM isolates were from Hausa and Ngas patients that belonged to the Afro-Asiatic language family. Monoresistance was more prevalent among Moslems (42.3% *vs.* 22.0%, P = 0.08) (S7 Table in S1 File), but the two religious groups did not differ significantly with regard to the genotype distribution (S9 Table in S1 File).

As for the age stratification, patients infected with the Cameroon lineage isolates had a mean age of 34.4±10.4 years (range 15–58), and those excreting T clonal line isolates had a mean age of 38.4±11.5 years (range 19–58). The proportion of young patients (≤35 years old) was higher in Cameroon than T group (55.8% *vs*. 36.8%, P = 0.17).

Males predominated in both Cameroon and T spoligotype lineages (80.9% and 78%).

## MIRU-VNTR typing

A total of 7 MIRU-VNTR types were described, shared by 2 to 17 isolates, totaling 43 (64.2%) isolates in clusters. The remaining 24 (35.8%) isolates harbored unique patterns. None of the VNTR types were associated with a particular ethnic group or place of living. The two largest clusters comprised of 17 and 11 isolates, representing different regions and two major language families, i.e. Niger-Congo and Afro-Asiatic (Fig 2).

## Combined analysis

A combined spoligotyping and MIRU-VNTR analysis resolved 53 (79.1%) unique patterns, and yielded a HGDI of 0.995. The HGDIs for spoligotyping and MIRU-VNTR alone were of 0.713 and 0.905, respectively. The clusters (6 in total) comprised a maximum of 3 isolates, producing the overall clustering rate of 20.9% and RTI of 11.9%. However, only one cluster (n = 2; 3%) contained isolates of the same resistance profile (STR-resistant) and recovered from patients of the same ethnicity (Hausa) and place of living (urban area in the northern part of the Plateau State).

## Discussion

### Association of genotypes with phenotypic and patient-related characteristics. TB transmission

In previous studies from Cameroon and Nigeria, male gender and age between 25 and 34 years were significantly associated with the Cameroon lineage [10, 14, 24]. Here, a proportion of younger patients was clearly higher among those infected by the Cameroon (55.8%) compared to the T lineage bacilli (36.8%). This difference was not significant, possibly due to a small sample size. Males predominated (77.6%), irrespective of the genotype.

No relationship was observed between the Cameroon clonal line and drug resistance in the previous studies [12, 14, 24]. In the present one, somewhat higher MDR rate was found for the T lineage compared to Cameroon, but the difference was insignificant. Thus, the increasing prevalence rate of the Cameroon clonal line in this and some previous studies cannot be attributed to its capacity to acquire MDR phenotype. In the first study that described this lineage, in Cameroon, it was noted that the clonal line was distributed homogeneously across the entire Ouest province and the reasons for its selection and dissemination were unknown [25]. The authors hypothesized the *M. bovis* BCG vaccination, which is common practice in Cameroon, may play a role in the selection of Cameroon lineage isolates. *In vivo* virulence studies may be useful to understand reasons behind dissemination the of this and other African genotypes.

Monoresistance was quite high (38.8%) and STR-monoresistance dominated (31.3%). However, STR-monoresistant isolates were located in different spoligotyping clusters (Fig 2). Furthermore, resistance to STR was not associated significantly with any patient characteristic. Thus, STR-monoresistant TB in the Plateau State is rather related to noncompliance or inadequate treatment than spread of a particular clone.

So far, only three studies have estimated the proportion of TB transmission in Nigeria, based on a combination of spoligotyping and MIRU-VNTR typing [10–12]. Using a 5-, 12-

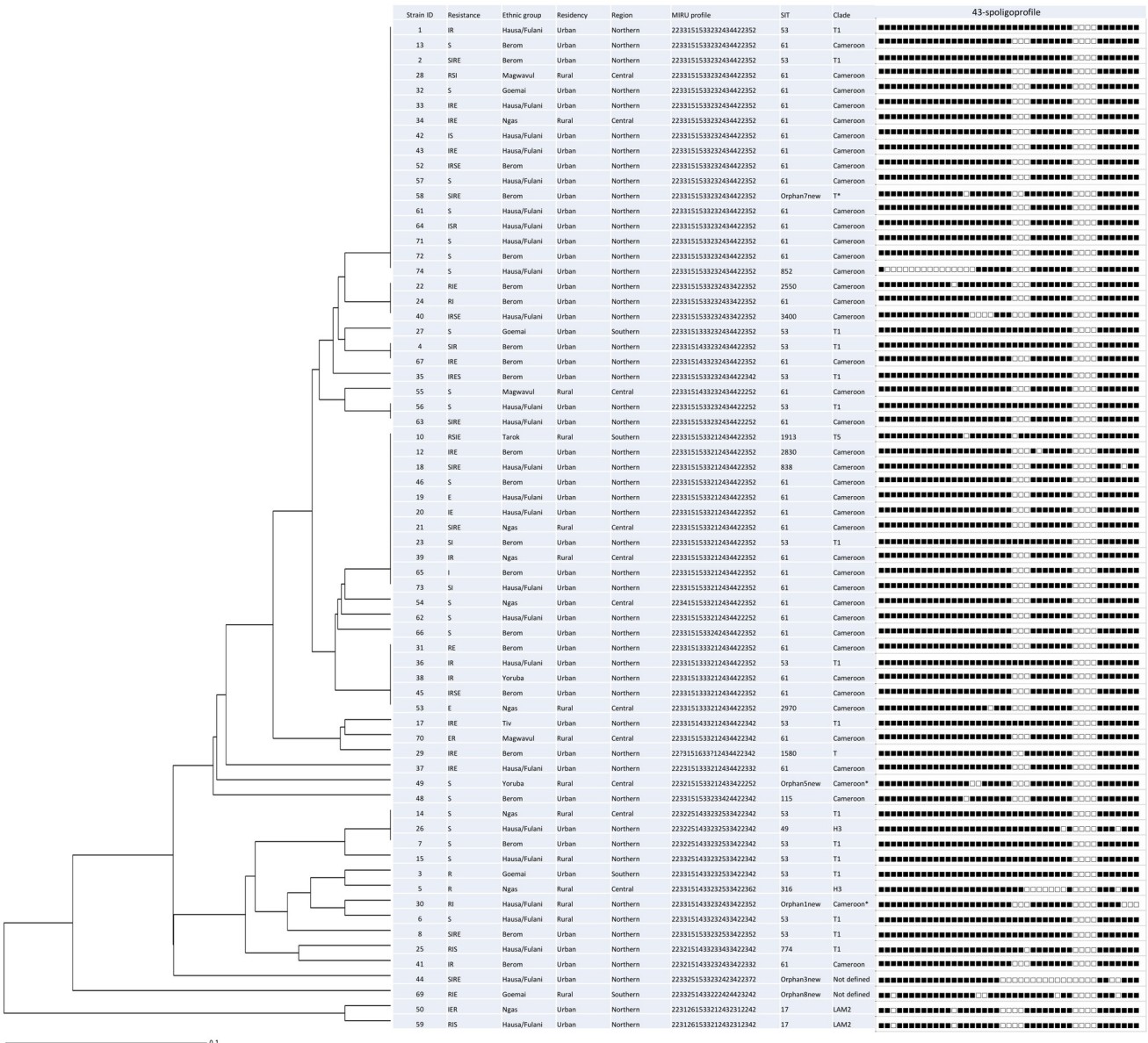

**Fig 2. Dendrogram based on the 22 VNTR loci for 67 *M. tuberculosis* isolates from the Plateau State, Nigeria.** Abbreviations: I–isoniazid; R–rifampicin; E–ethambutol; S–streptomycin; SIT—Shared International Type; 22 MIRU loci order: MIRU 2-4-10-16-20-23-24-26-27-31-39-40, Mt-04, ETR-C, Mt-21, ETR-A, Mt-29, Mt-30, ETR-B, Mt-34, QUB-26, QUB-4156c.

and 24-locus set, the RTI was calculated at 61.1%, 48.1%, and 38%, respectively. In this study, the RTI was importantly lower, (11.9%), which is even more surprising given the missing results from the two VNTR loci (22/24). In addition, clustered isolates were mostly isolated from patients of different socio-demographic characteristics and whose strains exhibited different resistance profiles. All this may indicate that patients developed TB due to reactivation of a latent infection or transmission in a distant past, rather than due to a recent transmission event. This might be somewhat surprising in the light of the patients' age (mean age 35.2±10.8

years). It thus seems that TB transmission in our group was somewhat underestimated, and that the criteria used for clustering, as a proxy for TB transmission, were too restrictive. As for the other countries in central/west Africa, the RTI reported, and established with the same methodology, was much higher, calculated at 42.6%, 35.8%, and 26.8% in Congo, Burkina Faso, and Benin, respectively [26–28]. Contrastingly, in a study from Cameroon, no MIR-U-VNTR clusters were observed, and thus no transmission was assumed [29].

## *M. tuberculosis* phylogeography in Nigeria and West Africa

While large-scale population structure is deeply rooted in time, the frequent everyday transborder movements can influence the spread of the particular clonal clusters and spoligotypes. To place our results in a wider phylogeographic context, we compared the main genotype lineages from this study with those observed in West Africa, and included dynamic changes, whenever available (Fig 3 and S10 Table in S1 File). The main SIT in this study was SIT61 (Cameroon lineage). In the first articles on *M. tuberculosis* strain typing in Africa it was noted that this predominant spoligotype in Cameroon was limited to West African countries (Benin, Senegal, and Ivory Coast) and to the Caribbean area [25]. SIT61 is the prototype of the Cameroon clonal line whose signature is deleted signals 23–25 and 33–36. The Cameroon lineage is one of the most known African genotypes of *M. tuberculosis*. Previously, based on spoligotyping, this clonal line was misnamed as LAM-CAM. More recently, MIRU-VNTR and WGS analysis demonstrated that it is a separate clonal line within the large and heterogeneous Euro-American lineage. Phylogeographically, the Cameroon strains are circulating in different African countries, and have been found on other continents, due to cross border migration. In SITVIT2, 1,095 isolates are assigned to the Cameroon lineage and the most numerous (*ca.* 40%) are samples from Nigeria and Cameroon itself. An increase of the Cameroon clonal line has been reported over years in Chad [30, 31] and Benin [27, 32] (S10 Table in S1 File). Similarly, in Guinea, 132 (71.7%) strains were genotypically clustered, and the Cameroon lineage was detected to be spreading significantly faster (p<0.001) than the background rate of the whole spoligotype population [33]. Overall, our meta-analytical approach showed that the prevalence of the Cameroon lineage was high in Cameroon, Nigeria, Burkina Faso, Benin, and

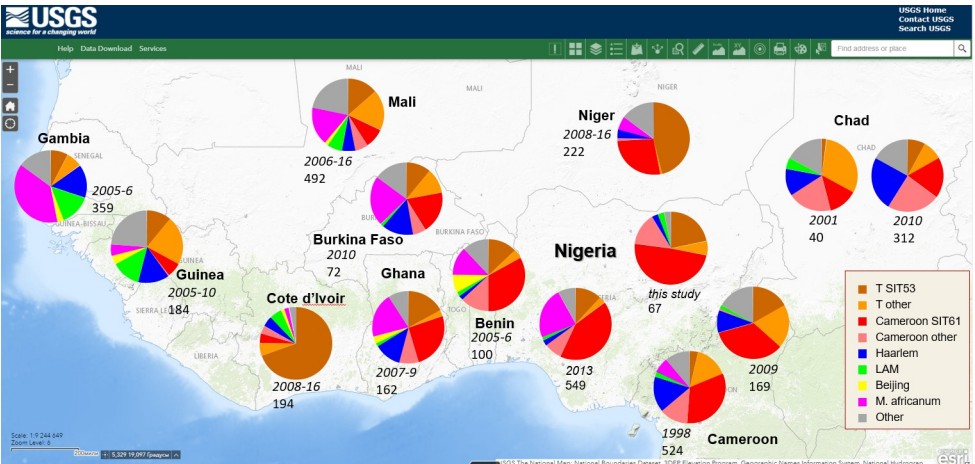

**Fig 3. Geographic distribution of the main *M. tuberculosis* spoligotype lineages in Nigeria and other parts of West Africa (see S10 Table in S1 File for details and references).** For each setting we show the years of surveillance and number of isolates studied. Free map: USGS National Map Viewer (public domain): http://viewer.nationalmap.gov/viewer/.

Ghana. The increasing trends for this clonal line were reported in Nigeria and Chad, while decreasing or low ratio in the western part of West Africa, e.g. in Ivory Coast, Gambia, Mali, and Guinea (Fig 3).

The second main SIT in this study was SIT53 (T lineage). According to our global analysis, SIT53 showed an increase in Cameroon and Chad, and a very high rate in Niger and Ghana (Fig 3). However, one should keep in mind that SIT53 is a heterogeneous group of distantly related strains otherwise found on different branches of the L4 tree based on genome-wide SNPs [34]. The SIT53 populations from diverse geographical locations most likely represent unrelated branches that evolved from ancestral SIT53 over historically long time [34]. In this sense, SIT53 populations in different parts of Africa may represent unrelated genotypes.

Minor groups in this study represented well-known global LAM and Haarlem clades. SIT17 belongs to the RD-Rio sublineage of the LAM clade and it was hypothesized that this genotype derived from SIT20 by single spacer deletion, originated in South America [35]. The published MIRU-VNTR-based phylogenetic analysis of the LAM lineage demonstrated that SIT17 isolates were present in Brazil, Venezuela, and Sierra-Leone. Interestingly, SIT17 was also described in Cameroon 20 years ago [25] and the identification of SIT17 in the present study could reflect its endemic circulation, albeit at a low rate. The same may be true for the Haarlem SIT49 spoligo-type that was also described in other country-wide study from Nigeria (5/549 isolates [13]), a study from Ivory Coast (2/194 isolates; [36]), and from Burkina Faso (1/72 isolates; [37]).

The *M. africanum* isolates were not included in this work. However, the prevalence of *M. africanum* has been decreasing in recent decades due to its reduced virulence [38]. According to previous reports from Nigeria, *M. africanum* was detected with a prevalence of 33% and 13% [10, 11]. More recently, only 3% of TB in Nigeria was shown to be due to *M. africanum* [14]. On the other hand, the low recovery of *M. africanum* might be explained by difficulties in isolating this species in routine practice [37].

Our own analysis of the geographic distribution of the causative agent of human TB in West Africa showed that the prevalence of *M. africanum* was low in Ivory Coast, Niger, and Guinea, and decreasing in Cameroon (Fig 3). Furthermore, *M. africanum* was not detected in Chad. In contrast, a high proportion of *M. africanum* was reported in Gambia, Mali, Burkina Faso, and Ghana, and in some parts of Nigeria (Fig 3). It appeared that there is no apparent pattern of *M. africanum* geographical distribution across West Africa. Thus, specific features of the national TB control programs and ethnic background may play a role in shaping the dissemination pattern of *M. africanum*.

The common problem with TB epidemiological studies is their small sample size. Thus, the distribution of *M. tuberculosis* lineages shown in Fig 3 may be just a random snapshot of lineage compositions in different countries. In this sense, Fig 3 presents the current state of knowledge that needs further studies.

Certain limitations of the study have to be mentioned. Firstly, study sample included 67 isolates originated from only one state of Nigeria. Therefore, the results could not be extrapolated to the entire country. Secondly, since only the conventional DST method was applied, the acquired genetic resistance could not be tracked under the study. Thirdly, a substantial improvement to the study would be a scrupulous clinical interview aiming at a better understanding of transmission routes of *M. tuberculosis* infection. Finally, of particular importance would be data concerning management and treatment outcomes of the patients.

## Conclusions

To conclude, this work describes the genetic diversity of DR *M. tuberculosis* strains circulating in the Plateau State, Nigeria, with a combination of spoligotyping and 22-loci MIRU-VNTR-

typing. There are three major findings from the study. First, all DR *M. tuberculosis* isolates were of the Euro-American lineage, and belonged to a limited number of spoligotype-defined lineages, mostly Cameroon and T clades (91%). The minor spoligotypes represented otherwise globally widespread Haarlem and LAM clonal lines. Second, the combination of spoligotyping and MIRU-VNTR typing resulted in a TB transmission rate of 11.9%, suggesting that recent transmission only scarcely contributes to the persistence of DR-TB in the Plateau State. Third, no association was observed between MDR or STR-monoresistance and Cameroon lineage. Thus, an increasing prevalence of this clonal line, observed here and in previous studies, cannot be associated with an increased acquisition of drug resistance. Further *in vivo* studies are needed to elucidate the pathobiological reasons behind the dissemination of *M. tuberculosis* lineages in Nigeria.

## Supporting information

**S1 File.**
(DOCX)

## Author Contributions

**Conceptualization:** Eugene I. Ikeh, Tomasz Jagielski.

**Funding acquisition:** Zofia Bakuła, Jacek Bielecki, Tomasz Jagielski.

**Investigation:** Zofia Bakuła, Valentine B. Wuyep, Łukasz Bartocha, Anna Vyazovaya.

**Methodology:** Zofia Bakuła, Valentine B. Wuyep, Anna Vyazovaya.

**Project administration:** Zofia Bakuła.

**Resources:** Jacek Bielecki.

**Supervision:** Eugene I. Ikeh, Igor Mokrousov, Tomasz Jagielski.

**Writing – original draft:** Zofia Bakuła, Igor Mokrousov.

**Writing – review & editing:** Igor Mokrousov, Tomasz Jagielski.

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
