## [Decision Letter · Decision Letter 0]

15 Feb 2022

PONE-D-21-26334Molecular snapshot of drug-resistant  Mycobacterium tuberculosis  strains from the Plateau State, NigeriaPLOS ONE

Dear Dr. Jagielski,

Thank you for submitting your manuscript to PLOS ONE. After careful consideration, we feel that it has merit but does not fully meet PLOS ONE’s publication criteria as it currently stands. Therefore, we invite you to submit a revised version of the manuscript that addresses the points raised during the review process.

Majority of reviewers have endorsed that it is a good study, but has some some major concerns, particularly the conclusions which should be based on hard data. I would suggest the authors to add a section of study limitation which should include beside other limitations, the phenotypic method (proportion method) of Drug resistance determination. 

We look forward to receiving your revised manuscript.

Kind regards,

Sarman Singh, MD, FRSC, FRCP

Academic Editor

PLOS ONE

https://journals.plos.org/plosone/s/fileid=ba62/PLOSOne_formatting_sample_title_authors_affiliations.pdf".

2. Please provide additional details regarding participant consent. In the ethics statement in the Methods and online submission information, please ensure that you have specified (1) whether consent was informed and (2) what type you obtained (for instance, written or verbal, and if verbal, how it was documented and witnessed). If the need for consent was waived by the ethics committee, please include this information.

“The study was financially supported by the Ministry of Science and Higher Education through the Faculty of Biology, University of Warsaw intramural grant DSM (501-D114-01-1140400).”

4. Thank you for stating the following in the Financing Section of your manuscript:

“The study was financially supported by the Ministry of Science and Higher Education through the Faculty of Biology, University of Warsaw intramural grant DSM (501-D114-01-1140400).”

“The study was financially supported by the Ministry of Science and Higher Education through the Faculty of Biology, University of Warsaw intramural grant DSM (501-D114-01-1140400).”

5. PLOS requires an ORCID iD for the corresponding author in Editorial Manager on papers submitted after December 6th, 2016. Please ensure that you have an ORCID iD and that it is validated in Editorial Manager. To do this, go to ‘Update my Information’ (in the upper left-hand corner of the main menu), and click on the Fetch/Validate link next to the ORCID field. This will take you to the ORCID site and allow you to create a new iD or authenticate a pre-existing iD in Editorial Manager. Please see the following video for instructions on linking an ORCID iD to your Editorial Manager account: https://www.youtube.com/watch?v=_xcclfuvtxQ".

6. We note that [Figure 3] in your submission contain [map/satellite] images which may be copyrighted. All PLOS content is published under the Creative Commons Attribution License (CC BY 4.0), which means that the manuscript, images, and Supporting Information files will be freely available online, and any third party is permitted to access, download, copy, distribute, and use these materials in any way, even commercially, with proper attribution. For these reasons, we cannot publish previously copyrighted maps or satellite images created using proprietary data, such as Google software (Google Maps, Street View, and Earth). For more information, see our copyright guidelines: http://journals.plos.org/plosone/s/licenses-and-copyright.

a. You may seek permission from the original copyright holder of Figure 3 to publish the content specifically under the CC BY 4.0 license. 

Reviewers' comments:

Reviewer's Responses to Questions

**Comments to the Author**

1. Is the manuscript technically sound, and do the data support the conclusions?

Reviewer #1: Yes

Reviewer #2: Yes

Reviewer #3: Partly

Reviewer #4: Yes

2. Has the statistical analysis been performed appropriately and rigorously? 

Reviewer #1: Yes

Reviewer #2: Yes

Reviewer #3: Yes

Reviewer #4: Yes

3. Have the authors made all data underlying the findings in their manuscript fully available?

Reviewer #1: Yes

Reviewer #2: Yes

Reviewer #3: Yes

Reviewer #4: Yes

4. Is the manuscript presented in an intelligible fashion and written in standard English?

Reviewer #1: Yes

Reviewer #2: Yes

Reviewer #3: Yes

Reviewer #4: Yes

5. Review Comments to the Author

Reviewer #1: The paper is about clonal composition of Mtb isolates from central Nigeria hospitals. For lineage identification, the authors used several standard genotyping techniques: spoligotyping and MIRU-VNTR-typing. Also, each isolate was characterized by drug susceptibility testing. Then the authors compared the Mtb lineage composition from Nigeria to those from other West African countries available from the literature.

The paper is interesting. The methods are adequate to the aims and the declared results of this study. I support publishing this paper after several minor corrections.

The common problem of all studies of this kind is the relatively small sample size. The authors had analyzed 67 Mtb isolates. Figure 3 looks for me as a random snapshot of Mtb lineage compositions in different countries due to this problem rather than because of any internal differences between countries in terms of disease distribution history, history of human migrations or any human genetic differences. The authors have to add several clear statements on whether they believe themselves into this picture on Fig. 3 and what can support this believe. Or if it is just the current stay of the art that needs further studies, please say it explicitly.

Throughout the text the authors use the term ‘families’ for Mtb lineages (lines 38, 54, 58, 64, 180, 182, 230, 292, 297, 299, 300, 314, 353, 354, 360. Better use the terms ‘lineages’, or ‘clonal lines’, or ‘genotypes’ to avoid confusing with taxonomic families (i.e. Mycobacteriaceae).

I am a bit confused with the male-to-female ratio of the patients. In line 155 it is said that the ratio was 3:5. But in line 230: Males predominated in both Cameroon and T spoligotype families (80.9% and 78%). And in line 259: Males predominated (77.6%), irrespective of the genotype. So, there were more female patients in the group but males dominated. Explain the statistics.

Line 71 – decipher MDR and RR-TB when these terms are used for the first time.

Line 89 – remove the dot before the reference in “…identified. [13].”

Reviewer #2: Bakuła et al. applied spoligotyping and MIRU-VNTR to establish the genetic diversity of drug-resistant M. tuberculosis strains circulating in the Plateau State in Nigeria. The number of Mtb isolates included in this study was quite limited (67 strains) but allowed to identify the transmission rate and the dominant lineages present in the investigated area. Based on spoligotyping data and MIRU-VNTR analysis Authors found that all drug-resistant isolates

in the Plateau State belong to Euro-American lineages, mostly Cameroon and T clades. In contradiction to the previous reports, the TB-transmission rate was determined as low as <12%.

The manuscript is well written. The weak points of the presented story are a low number of isolates included in the study and 22 (not 24) loci included in the MIRU-VNTR analysis (due to a technical reason). However, these weaknesses do not affect the conclusions made by the authors.

Comments:

Abstract – the sentence “Most of the patients have developed TB due to a reactivation of infection or from a transmission event in the distant past” is speculation and should be excluded from the Abstract section. The increased number of isolates included in the study could affect the transmission rate determined by the authors.

67 drug-resistant strains isolated in a period of three months were enrolled in the analysis. It should be mentioned what % is it of all Mtb isolates, and DR-Mtb isolates, in this area in the time of one year.

Fig. 2 should be replaced with the dendrogram prepared based on the combined analysis, spoligotyping, and MIRU-VNTR, or at least spoligotyping data (SIT) should be included in the dendrogram description.

Fig. 3 each pie chart located in Fig. 3 should be described with a number of isolates (n=x).

Reviewer #3: The work entitled: Molecular snapshot of drug-resistant Mycobacterium tuberculosis strains from the Plateau State, Nigeria by Bakula et al., 2022 is intended the provide information about the molecular epidemiology of M tuberculosis strains in Nigeria based pn DR Mtb strains placed in a the West Africa context. Nigeria ranks 1st in Africa and 6th with the highest burden of TB. Spoligo and MLVA typing were used to disclosure Mtb clustered strains (20.9%) and recent transmission rate was estimated at 11,9%. Most of the strains were represented by two Clades (91%), Cameroon and T. Interestingly, TB cases came from reactivation of previous infection and no association was detected between Cameroon Clade and drug resistance.

Below, my comments.

I believe that results should be presented together instead of separately. I mean, a table with drug susceptibility profiles + both molecular epidemiologic markers + origins + ethnicities + languages + religions. I may suggest to show subtotals of Non MDR and MDR strains in Table 1. Both molecular markers complement each other and hence produce reliable results defining real clusters and correcting lineage origins plus having a global view of the epidemiologic situation. It is well know that both molecular markers are prone to convergent evolution, and Spoligotyping is more susceptible. In this way reading will be much more easy and analysis will be more afordable.

For example, in line 199 you say that two Haarlem strains had different SITs (Table 2), but they could be linked directly to each other with one deletion event….. it is well known that unique deletion events could not be linked with spoligo data alone and that a second and more robust marker is needed to verify this situation due to convergent evolution. What I see is a block of spacers deleted at the left side of the Haarlem spoligo signature, an event that is not possible to know if happened in a single step. I suggest to re-write this sentence.

It would have been highly appreciated a dendogram with all data suggested previously to be presented in one Table. This provides the most important and reliable information.

In page 11, line 230 you say that males predominate in Cameroon and T families and in line 259 (page 12) again irrespective of the genotype but previously (page 7, line 155) you said that male:female ratio was 3:5? could you please explain better what you really want to say.

Phrase in line 268 should be revisited for English writing.

You guys say that TB was developed due to reactivation of a previous infection rather than from a recent transmission event, based mainly on the low RTI and the clustering rate by epimol tools. However, sample size and the criteria to be included in the present study influence this low rate observed. I would like to add that your results show that the mean age of 67 enrolled patients was 35.2±10.8 years (range 15-58). When TB is present in patients around this age it is believed that should have been the result of TB transmission more than reactivation, that is expected in older people.

The biggest cluster, involving 17 strains, show that most of the strains came from the Northen and urban region of Nigeria. Strains 1, 2, 28, 33, 34, 43, 52, 58, 64 share at least resistance to IR and also they share the Hausa/Fulani ethnicity (Fig 2). These strains share a link plus the VNTR profile hence they might be transmitting in the population at a low rate. The arisen question is how is adherence to treatment?

The lack of treatment adherence selects for drug resistance appearance hence some patients that became infected with the transmitting susceptible strain will have the same strain type but now will became resistant. Classic molecular epidemiology surveys are needed to clarify this situation.

We must remember that drug susceptibility testing was based on proportion method rather than mutation detection which means that a population of bacilli could be evolving to develop resistance to any drug but may not be detected due to the test sensitivity unless the mutation conferring resistance will be prevalent in patient´s bacilli. It is well known that drug resistant strains mainly transmit at a lower rate than susceptible strains with rare exceptions. I conclusion, it may not be exactly the same strain but a very related one that arose by evolution from the parental strain.

I may suggest to say something about this.

Data supports partlaly the conclusions in the way they are written.

You should mention in Method which Statistical method you used.

All Data is avaiable but should be better presented.

The manuscript was wrote in standard english however some parts must be re-written.

Reviewer #4: This study is well written and provides interesting TB molecular epidemiology analyis in Nigeria.

However, I have some comments to improve it.

In the abstract section (line 31), you mentioned "a total of 20 spoligotypes", is this number corresponding to distinct spoligotypes ?

On line 29, you said "The study sample included 67 DR M. tuberculosis isolates"

Cote d’Ivoire is not always well written in the article.

6. PLOS authors have the option to publish the peer review history of their article (what does this mean?). If published, this will include your full peer review and any attached files.

Reviewer #1: **Yes: **Oleg N. Reva, Dep. Biochemistry, Genetics and Microbiology; Centre for Bioinformatics and Computational Biology, University of Pretoria, Pretoria, South Africa.

Reviewer #2: No

Reviewer #3: No

Reviewer #4: No

---

## [Author Response · Author response to Decision Letter 0]

10 Mar 2022

Warsaw,

21st February, 2022

To:

Sarman Singh, MD, FRSC, FRCP

Academic Editor

PLOS ONE

Dear Sarman Singh,

I would like to thank you, as well as your expert Reviewers, for helping us to improve our manuscript (PONE-D-21-26334). You will find below our answers (highlighted in grey).

Editor:

1. Majority of reviewers have endorsed that it is a good study, but has some major concerns, particularly the conclusions which should be based on hard data. I would suggest the authors to add a section of study limitation which should include beside other limitations, the phenotypic method (proportion method) of Drug resistance determination. 

We appreciate a positive feedback from the Editor. Now we have added section regarding study limitations in the “Discussion”. Please see below.

“Certain limitations of the study have to be mentioned. Firstly, study sample included 67 isolates originated from only one state of Nigeria. Therefore, the results could not be extrapolated to the entire country. Secondly, since only the conventional DST method was applied, the acquired genetic resistance could not be tracked under the study. Thirdly, a substantial improvement to the study would be a scrupulous clinical interview aiming at a better understanding of transmission routes of M. tuberculosis infection. Finally, of particular importance would be data concerning management and treatment outcomes of the patients.”

Reviewer #1: The paper is about clonal composition of Mtb isolates from central Nigeria hospitals. For lineage identification, the authors used several standard genotyping techniques: spoligotyping and MIRU-VNTR-typing. Also, each isolate was characterized by drug susceptibility testing. Then the authors compared the Mtb lineage composition from Nigeria to those from other West African countries available from the literature.

The paper is interesting. The methods are adequate to the aims and the declared results of this study. I support publishing this paper after several minor corrections.

We would like to thank the Reviewer for this positive comment. 

1. The common problem of all studies of this kind is the relatively small sample size. The authors had analyzed 67 Mtb isolates. Figure 3 looks for me as a random snapshot of Mtb lineage compositions in different countries due to this problem rather than because of any internal differences between countries in terms of disease distribution history, history of human migrations or any human genetic differences. The authors have to add several clear statements on whether they believe themselves into this picture on Fig. 3 and what can support this believe. Or if it is just the current stay of the art that needs further studies, please say it explicitly.

We agree with this comment. However some of the studies shown in Fig. 3 were not so small (N>300-500). The following paragraph was added to the revised version of the manuscript in the “Discussion” section. Please see below. 

"The common problem with TB epidemiological studies is their small sample size. Thus, the distribution of M. tuberculosis lineages shown in Fig. 3 may be just a random snapshot of lineage compositions in different countries. In this sense, Fig. 3 presents the current state of knowledge that needs further studies.”

2. Throughout the text the authors use the term ‘families’ for Mtb lineages (lines 38, 54, 58, 64, 180, 182, 230, 292, 297, 299, 300, 314, 353, 354, 360. Better use the terms ‘lineages’, or ‘clonal lines’, or ‘genotypes’ to avoid confusing with taxonomic families (i.e. Mycobacteriaceae).

The suggested correction has been made in the revised manuscript and supplementary tables. 

3. I am a bit confused with the male-to-female ratio of the patients. In line 155 it is said that the ratio was 3:5. But in line 230: Males predominated in both Cameroon and T spoligotype families (80.9% and 78%). And in line 259: Males predominated (77.6%), irrespective of the genotype. So, there were more female patients in the group but males dominated. Explain the statistics.

We thank the Reviewer for pointing out this error. In the study there were 15 females (22.4%), and 42 males (77.6%). We have now corrected this mistake, as follows (changes are marked in bold).

 “Male-to-female ratio was ca. 4:1.”

4. Line 71 – decipher MDR and RR-TB when these terms are used for the first time.

The correction has been made.

5. Line 89 – remove the dot before the reference in “…identified. [13].”

We have corrected this.

Reviewer #2: Bakuła et al. applied spoligotyping and MIRU-VNTR to establish the genetic diversity of drug-resistant M. tuberculosis strains circulating in the Plateau State in Nigeria. The number of Mtb isolates included in this study was quite limited (67 strains) but allowed to identify the transmission rate and the dominant lineages present in the investigated area. Based on spoligotyping data and MIRU-VNTR analysis Authors found that all drug-resistant isolates in the Plateau State belong to Euro-American lineages, mostly Cameroon and T clades. In contradiction to the previous reports, the TB-transmission rate was determined as low as <12%.

The manuscript is well written. The weak points of the presented story are a low number of isolates included in the study and 22 (not 24) loci included in the MIRU-VNTR analysis (due to a technical reason). However, these weaknesses do not affect the conclusions made by the authors.

We appreciate the Reviewer's positive feedback

1. Abstract – the sentence “Most of the patients have developed TB due to a reactivation of infection or from a transmission event in the distant past” is speculation and should be excluded from the Abstract section. The increased number of isolates included in the study could affect the transmission rate determined by the authors.

As suggested by the Reviewer, we have now removed this sentence from the “Abstract”. 

2. 67 drug-resistant strains isolated in a period of three months were enrolled in the analysis. It should be mentioned what % is it of all Mtb isolates, and DR-Mtb isolates, in this area in the time of one year.

We agree with the Reviewer, that data regarding percentage of all M. tuberculosis isolates collected in the sampled area (and during the sampling period), along with isolates’ drug resistance profiles, would be of much value for the study. However, the data on the total number of M. tuberculosis isolates collected in the study period from the Ministry of Health TB Prevention and Control Center in Jos were unavailable. Furthermore, drug resistance profiles were unavailable for most of the isolates. Thus, initially we decided not to include those data in the manuscript. 

We have now commented on this in the “Materials and Methods” section. Please see below.

“The total number of M. tuberculosis isolates collected in the study period was 45 for the Plateau State Specialist Hospital TB DOTs Center and 145 for Jos University Teaching Hospital MDR-TB Clinic. Data on the total M. tuberculosis isolates from the Ministry of Health TB Prevention and Control Center in Jos was unavailable. Also, not for all M. tuberculosis isolates were data on their drug resistance profiles obtainable.”

3. Fig. 2 should be replaced with the dendrogram prepared based on the combined analysis, spoligotyping, and MIRU-VNTR, or at least spoligotyping data (SIT) should be included in the dendrogram description.

The suggested correction has been made. Please see Fig. 2.

4. Fig. 3 each pie chart located in Fig. 3 should be described with a number of isolates (n=x).

The suggested correction has been made. Please see Fig. 3.

Reviewer #3: The work entitled: Molecular snapshot of drug-resistant Mycobacterium tuberculosis strains from the Plateau State, Nigeria by Bakula et al., 2022 is intended the provide information about the molecular epidemiology of M tuberculosis strains in Nigeria based on DR Mtb strains placed in a the West Africa context. Nigeria ranks 1st in Africa and 6th with the highest burden of TB. Spoligo and MLVA typing were used to disclosure Mtb clustered strains (20.9%) and recent transmission rate was estimated at 11,9%. Most of the strains were represented by two Clades (91%), Cameroon and T. Interestingly, TB cases came from reactivation of previous infection and no association was detected between Cameroon Clade and drug resistance.

Below, my comments.

I believe that results should be presented together instead of separately. I mean, a table with drug susceptibility profiles + both molecular epidemiologic markers + origins + ethnicities + languages + religions. I may suggest to show subtotals of Non MDR and MDR strains in Table 1. Both molecular markers complement each other and hence produce reliable results defining real clusters and correcting lineage origins plus having a global view of the epidemiologic situation. It is well know that both molecular markers are prone to convergent evolution, and Spoligotyping is more susceptible. In this way reading will be much more easy and analysis will be more afordable.

As suggested by the Reviewer 1, we have incorporated spoligotyping and MIRU-VNTR typing analysis into Figure 2. Therefore, now all the data are presented together in one figure. Furthermore, we have added the subtotals of non MDR and MDR strains in Table 1, as requested. 

1. For example, in line 199 you say that two Haarlem strains had different SITs (Table 2), but they could be linked directly to each other with one deletion event….. it is well known that unique deletion events could not be linked with spoligo data alone and that a second and more robust marker is needed to verify this situation due to convergent evolution. What I see is a block of spacers deleted at the left side of the Haarlem spoligo signature, an event that is not possible to know if happened in a single step. I suggest to re-write this sentence.

As requested by the Reviewer, we have rephrased the sentence. Please see below.

“Two Haarlem strains had different SITs (Table 2) and due to their single deletion event difference, they could not be directly linked to each other.”

It would have been highly appreciated a dendogram with all data suggested previously to be presented in one Table. This provides the most important and reliable information.

We have already addressed this issue. Now all the data are combined in Figure 2. If requested, we can add Supplementary Excel File, with all the raw data. 

In page 11, line 230 you say that males predominate in Cameroon and T families and in line 259 (page 12) again irrespective of the genotype but previously (page 7, line 155) you said that male:female ratio was 3:5? could you please explain better what you really want to say.

Male to female ratio was wrongly calculated in the manuscript. We are sorry about it. The study included 15 females (22.4%), and 42 males (77.6%). Therefore, the predominance of males in the Cameroon and T families is not surprising. 

We have now corrected the error, as follows (changes are marked in bold).

“Male-to-female ratio was ca. 4:1.”

Phrase in line 268 should be revisited for English writing.

We thank the Reviewer for this comment. We have now reparsed the sentence (see below): 

“The authors hypothesized the M. bovis BCG vaccination, which is common practice in Cameroon, may play a role in the selection of Cameroon lineage isolates.”

You guys say that TB was developed due to reactivation of a previous infection rather than from a recent transmission event, based mainly on the low RTI and the clustering rate by epimol tools. However, sample size and the criteria to be included in the present study influence this low rate observed. I would like to add that your results show that the mean age of 67 enrolled patients was 35.2±10.8 years (range 15-58). When TB is present in patients around this age it is believed that should have been the result of TB transmission more than reactivation, that is expected in older people.

We thank the Reviewer for this valuable comment. We have now rephrased the Discussion section. Please see below (changes are marked in bold).

“All this may indicate that patients developed TB due to reactivation of a latent infection or transmission in a distant past, rather than due to a recent transmission event. This might be somewhat surprising in the light of the patients’ age (mean age 35.2±10.8 years). It thus seems that TB transmission in our group was somewhat underestimated, and that the criteria used for clustering, as a proxy for TB transmission, were too restrictive.”

The biggest cluster, involving 17 strains, show that most of the strains came from the Northern and urban region of Nigeria. Strains 1, 2, 28, 33, 34, 43, 52, 58, 64 share at least resistance to IR and also they share the Hausa/Fulani ethnicity (Fig 2). These strains share a link plus the VNTR profile hence they might be transmitting in the population at a low rate. The arisen question is how is adherence to treatment?

We agree with the Reviewer that data regarding history of treatment is of great importance in studies like ours. Sadly, data on patients’ clinical management were either fragmentary (precluding reconstruction of treatment history) or completely absent. We have added now the following sentences in the “Materials and Methods” and “Discussion” section.

“Since no data on patient management and long-term treatment outcomes were available, they were excluded from the analysis.”

“Finally, of particular importance would be data concerning management and treatment outcomes of patients.”

The lack of treatment adherence selects for drug resistance appearance hence some patients that became infected with the transmitting susceptible strain will have the same strain type but now will became resistant. Classic molecular epidemiology surveys are needed to clarify this situation. We must remember that drug susceptibility testing was based on proportion method rather than mutation detection which means that a population of bacilli could be evolving to develop resistance to any drug but may not be detected due to the test sensitivity unless the mutation conferring resistance will be prevalent in patient´s bacilli. It is well known that drug resistant strains mainly transmit at a lower rate than susceptible strains with rare exceptions. I conclusion, it may not be exactly the same strain but a very related one that arose by evolution from the parental strain. I may suggest to say something about this.

As suggested by the Reviewer, we have now commented on this in the paragraph regarding study limitations in the “Discussion” section. Please see below:

“Secondly, since only the conventional DST method was applied, the acquired genetic resistance could not be tracked under the study”

Data supports partially the conclusions in the way they are written. You should mention in Method which Statistical method you used. All Data is avaiable but should be better presented.

The manuscript was wrote in standard english however some parts must be re-written.

We thank the Reviewer for these concluding remarks. We have addressed them carefully, as described above. Furthermore, the description of statistical methods used in this study is depicted in the Material and Methods section. Please see below.

“A chi-square test was used to detect any significant difference between the two groups. Yates corrected χ2 and P-values were calculated with 95% confidence interval at MEDCALC (http://www.medcalc.org/calc/odds_ratio.php) online resource.”

Reviewer #4: This study is well written and provides interesting TB molecular epidemiology analyis in Nigeria.

However, I have some comments to improve it.

We would like to thank the Reviewer for this positive comment. 

In the abstract section (line 31), you mentioned "a total of 20 spoligotypes", is this number corresponding to distinct spoligotypes ?

Yes. This number corresponds to distinct spoligotype patterns. We have rephrased the sentence for better clarity. Please see below (changes are marked in bold).

“A total of 20 distinct spoligotypes were obtained, split into 3 clusters (n=50, 74.6%, 2-33 isolates per cluster) and 17 (25.4%) unique patterns”.

On line 29, you said "The study sample included 67 DR M. tuberculosis isolates"

As stated in the Materials and Methods section, the study sample included 67 M. tuberculosis isolates, resistant to at least one first-line drug. Therefore, the sentence “The study sample included 67 DR M. tuberculosis isolates" is correct. 

Cote d’Ivoire is not always well written in the article.

We thank the Reviewer for bringing our notice to that. We have now corrected the “Ivory Coast” name through the manuscript.

---

## [Editor Report · Decision Letter 1]

29 Mar 2022

Molecular snapshot of drug-resistant  Mycobacterium tuberculosis  strains from the Plateau State, Nigeria

PONE-D-21-26334R1

Dear Dr. Jagielski,

We’re pleased to inform you that your manuscript has been judged scientifically suitable for publication and will be formally accepted for publication once it meets all outstanding technical requirements.

Kind regards,

Sarman Singh, MD, FRSC, FRCP

Academic Editor

PLOS ONE

---

## [Editor Report · Acceptance letter]

12 May 2022

PONE-D-21-26334R1 

Molecular snapshot of drug-resistant *Mycobacterium tuberculosis* strains from the Plateau State, Nigeria 

Dear Dr. Jagielski:

I'm pleased to inform you that your manuscript has been deemed suitable for publication in PLOS ONE. Congratulations! Your manuscript is now with our production department. 

Kind regards, 

on behalf of

Professor Sarman Singh 

Academic Editor

PLOS ONE